

# Estimation of bubbled-mediated air/sea gas exchange from concurrent DMS and CO$_2$ transfer velocities at intermediate-high wind speeds

**Thomas G. Bell[1]**[*], **Sebastian Landwehr[2]**, **Scott D. Miller[3]**, **Warren J. de Bruyn[4]**, **Adrian Callaghan[5]**, **Brian Scanlon[2]**, **Brian Ward[2]**, **Mingxi Yang[1]** and **Eric S. Saltzman[6]**

[1] Plymouth Marine Laboratory, Prospect Place, The Hoe, Plymouth, PL1 3DH, UK

[2] School of Physics, National University of Ireland, Galway, Ireland

[3] Atmospheric Sciences Research Center, State University of New York at Albany, NY, USA

[4] Schmid College of Science and Technology, Chapman University, Orange, California, CA, USA

[5] Scripps Institution of Oceanography, University of California San Diego, 9500 Gilman Drive, La Jolla, CA 92093

[6] Department of Earth System Science, University of California, Irvine, CA, USA

*Correspondence to: T.G. Bell (tbe@pml.ac.uk)

## Abstract

Simultaneous air/sea fluxes and concentration differences of dimethylsulfide (DMS) and carbon dioxide (CO$_2$) were measured during a summertime North Atlantic cruise in 2011. This dataset reveals significant differences between the gas transfer velocities of these two gases ($\Delta k_w$) over a range of wind speeds up to 21 m s$^{-1}$. These differences occur at and above the approximate wind speed threshold when waves begin breaking. Whitecap fraction (a proxy for bubbles) was also measured and has a positive relationship with $\Delta k_w$, consistent with enhanced bubble-mediated transfer of the less soluble CO$_2$ relative to that of the more soluble DMS. However, the correlation of $\Delta k_w$ with whitecap fraction is no stronger than with wind speed. Models used to estimate bubble-mediated transfer from *in situ* whitecap fraction under-predict the observations, particularly at intermediate wind speeds. Examining the differences between gas transfer velocities of gases with different solubilities is a useful way to detect the impact of bubble-mediated exchange. More simultaneous gas transfer




measurements of different solubility gases across a wide range of oceanic conditions are
needed to understand the factors controlling the magnitude and scaling of bubble-mediated
gas exchange.

## 1  Introduction

Air/sea exchange is a significant process for many compounds that have biogeochemical and
climatic importance.  Approximately 25% of the carbon dioxide ($CO_2$) released into the
atmosphere by anthropogenic activities has been taken up by the world oceans, which has
tempered its climate forcing while leading to ocean acidification (Le Quéré et al., 2015).  The
biogenic gas dimethylsulfide (DMS) is a major contributor to the mass of marine atmospheric
aerosol (Virkkula et al., 2006). Volatile organic compounds (VOCs) such as isoprene,
acetone and acetaldehyde alter the oxidising capacity of the troposphere (Carpenter et al.,
2012). The solubility differences between these VOCs mean that their exchange is controlled
to differing degrees by processes on the water and air side of the air/sea interface (Yang et al.,
2014). Many of the factors influencing air/sea gas exchange will be altered by future changes
in climate, ocean circulation and biology. Earth system models and air quality models require
more accurate understanding of the processes that influence air/sea gas transfer.
Air/sea gas exchange is typically parameterised as a function of the ocean/atmosphere bulk
concentration difference ($\Delta C$) and the physical mixing induced by wind stress at the interface
(Liss and Slater, 1974).  The air/sea flux is typically described using the expression:

$$Flux = K(C_w - \alpha C_a) \qquad \text{Equation 1}$$

where $C_w$ and $C_a$ are the trace gas bulk concentration on either side of the interface, $\alpha$ is the
dimensionless water/air solubility of the gas in seawater and $K$ is the gas transfer velocity.
The physics of gas transfer are implicitly represented by the gas transfer velocity, which is
commonly expressed in water-side units of velocity (cm hr$^{-1}$) and parameterized as a function
of wind speed ($U_{10}$) and Schmidt number ($Sc$). The simplicity of Equation 1 belies the
complexity of the processes involved in air/sea gas transfer.  These processes include
diffusion, surface renewal, buoyancy effects, wave-induced mixing, wave breaking and
bubble-mediated transport.  A variety of theoretical, laboratory, and field approaches have
been used to study these processes but we do not yet have a firm understanding of factors that
control air/sea transfer under a range of oceanic conditions.



The gas transfer-wind speed relationships for gases of different solubility may be affected by
breaking waves and bubbles (Woolf, 1993; Keeling, 1993; Woolf, 1997). Gas transfer via
bubbles ($k_{bub}$) is sensitive to the void fraction (ratio of air volume to total volume) of the
bubble plume as well as the bubble size distribution. Bubble injection depth and cleanliness
of the surface (influenced by surfactants) affect bubble rise velocity and residence time.
Bubble residence time determines the time available for equilibration to occur while bubble
volume and gas diffusivity ($Sc$) govern the time needed for a bubble to equilibrate. The
magnitude of $k_{bub}$ is expected to be greater for sparingly soluble gases (e.g. $CO_2$,
dimensionless solubility ~1) than for more soluble gases such as DMS (dimensionless
solubility ~15), particularly when bubbles are fully equilibrated.   Bubble-mediated gas
transfer has been studied in the laboratory (Rhee et al., 2007; Asher et al., 1996) and using
models (e.g. Woolf, 2005; Fairall et al., 2011; Woolf et al., 2007; Goddijn-Murphy et al.,

71   2016).

Deliberate, dual-tracer techniques have estimated gas transfer by measuring the evasion of a
pair of sparingly soluble gases with different diffusivity ($^3$He and $SF_6$, dimensionless
solubility $\leq$0.01). These studies indicate non-linear wind speed dependence of the gas transfer
velocity, in qualitative agreement with earlier studies in wind-wave tanks (e.g. Watson et al.,
1991; Wanninkhof et al., 1985; Liss and Merlivat, 1986).  Direct, shipboard measurements of
waterside gas transfer have also been made by eddy covariance (e.g. Bell et al., 2013;
Marandino et al., 2007; McGillis et al., 2001; Miller et al., 2010; Huebert et al., 2004).  These
measurements typically show DMS gas transfer velocities that are lower and exhibit more
linear wind speed dependence than those estimated for $CO_2$ based on dual tracer studies (e.g.
Bell et al., 2015; Yang et al., 2011; Goddijn-Murphy et al., 2012).  It has been suggested that
the difference between the open ocean gas transfer velocities of $CO_2$ and DMS is due to the
reduced importance of bubble-mediated exchange for DMS (Goddijn-Murphy et al., 2016;
Blomquist et al., 2006; Fairall et al., 2011).
Only one set of concurrent $CO_2$ and DMS gas transfer velocity measurements have been
published to date (Miller et al., 2009). In that study, no statistically significant difference was
observed in gas transfer-wind speed relationships of $CO_2$ and DMS for winds below 10 m s$^{-1}$.
This study presents a more extensive set of $CO_2$ and DMS gas transfer velocities that were
measured simultaneously aboard the R/V Knorr in the 2011 summertime North Atlantic in
both oligotrophic and highly productive waters.  The DMS and $CO_2$ gas transfer velocities





are discussed separately in detail by Bell et al. (2013) and Miller et al., In Prep.  Here we
focus specifically on what can be learned about gas transfer from the differences in behaviour
of two different solubility gases at intermediate and high wind speeds.
**2    Methods**
**2.1    Seawater, atmospheric and flux measurement systems**
The measurement setups for DMS and $CO_2$ concentrations in air and water and the eddy
covariance flux systems have been discussed in detail elsewhere (Bell et al., 2015; Bell et al.,
2013; Miller et al., 2010; Miller et al., 2008; Landwehr et al., 2014; Landwehr et al., 2015;
Saltzman et al., 2009).  We provide a summary and some additional details in the Appendix
(Section 6).
**2.2    Gas transfer velocity calculations**
In this section we describe the calculation of DMS and $CO_2$ gas transfer velocities from the
Knorr_11 cruise data.  Measured gas transfer velocities are transformed into water side only
gas transfer velocities in order to remove the influence of air-side resistance. Air-side
resistance is a function of solubility and thus different for the two gases. Finally, we discuss
the most appropriate approach for comparing the water-side gas transfer velocities, given that
the two gases have different molecular diffusivity and solubility.
Total gas transfer velocities ($K$) are calculated for $CO_2$ and DMS for each 10-minute flux
interval of the Knorr_11 cruise using Equation 1. The temperature-dependent dimensionless
solubility of $CO_2$ and DMS in seawater is calculated following Weiss (1974) and Dacey et al.
(1984). These gas transfer velocities reflect the result of resistance on both sides of the
interface (Liss and Slater, 1974). The water side contribution to the total resistance is
determined as follows:
$$k_w = \left[ \frac{1}{K} - \frac{\alpha}{k_a} \right]^{-1} \qquad\qquad Equation\ 2$$
where $k_w$ and $k_a$ are the air side and water side gas transfer velocities and $\alpha$ is dimensionless
water/air solubility. Note that we use the $\alpha$ reported by Dacey et al. (1984) in these
calculations rather than $H$ as there appears to be an error in conversion between $\alpha$ and  H in
that study (see Supplemental information for discussion). $CO_2$ solubility is sufficiently low





that air side resistance is negligible and the water side gas transfer is assumed equal to the
total transfer velocity ($k_{CO_2} = K_{CO_2}$). The air side resistance for DMS needs to be accounted
for because it is a moderately soluble gas (McGillis et al., 2000).  Air side gas transfer
velocities ($k_a$) for DMS were calculated for each 10 minute flux interval with the NOAA
COAREG 3.1 model, using sea surface temperature (SST) and horizontal wind speed
measured during the cruise. The NOAA COAREG 3.1 model (Fairall et al., 2011) is an
extension of the COARE bulk parameterization for air/sea energy and momentum fluxes to
simulate gas transfer (Fairall et al., 1998; Fairall et al., 2000).  The air side gas transfer
contributes about 5% on average to the total resistance for DMS.  NOAA COAREG 3.1
model calculations were carried out using a turbulent/molecular coefficient, A = 1.6, and
bubble-mediated coefficient, B = 1.8 (Fairall et al., 2011).  Knorr_11 measurements of SST,
air temperature, relative humidity, air pressure, downward radiation and wind speed were
used as input parameters to the model.
To facilitate comparison of transfer coefficients for the two gases across a range of sea
surface temperatures, gas transfer velocities are corrected for changes in molecular diffusivity
and viscosity. The correction typically involves the normalisation of water side gas transfer
velocities to a common Schmidt number ($Sc$=660), equivalent to $CO_2$ in seawater at 20°C:

$$k_{X,660} = k_X \cdot \left( \frac{660}{Sc_X} \right)^{-0.5} \qquad\qquad \textit{Equation 3}$$

where subscript $_X$ refers to $CO_2$ or DMS (i.e. $k_{DMS,660}$ and $k_{CO_2,660}$). Temperature-dependent
$Sc_{CO2}$ and $Sc_{DMS}$ were obtained using the *in situ* seawater temperature from the ship's bow
sensor and parameterisations from Wanninkhof (1992) and Saltzman et al. (1993).
The $Sc$ number normalization (Equation 4) is commonly used across the whole range of wind
speeds.  In fact, it is appropriate only for low or moderate winds in which interfacial gas
transfer dominates over bubble-mediated gas exchange.  If bubbles are an important
component of gas transfer then solubility also plays a role and normalization based on $Sc$
alone may not be sufficient.
To develop a more rigorous comparison of $k_{DMS}$ and $k_{CO_2}$, we normalized the water side
transfer velocities of DMS to the Schmidt number of $CO_2$ at the *in situ* sea surface
temperature of each 10-minute flux interval, as follows:



$$k_{DMS,Sc} = k_{DMS} \cdot \left( \frac{Sc_{CO_2}}{Sc_{DMS}} \right)^{-0.5}$$
*Equation 4*

where $Sc_{CO_2}$ and $Sc_{DMS}$ are the Schmidt numbers of $CO_2$ and DMS at the *in situ* sea surface
temperature. Compared to normalizing both DMS and $CO_2$ to Sc=660, this approach has the
advantage of correcting only $k_{DMS}$, with no correction to $k_{CO2}$. The *Sc* correction for DMS
should be reasonably accurate, assuming that the bubble-mediated transfer for the more
soluble DMS is relatively small.
On the Knorr_11 cruise, the variability in sea surface temperature was small ($1\sigma = \pm1°C$). As
a result, there is little difference in the variability or wind speed dependence of *Sc*-corrected
$k_{CO2}$ compared to $k_{CO2}$ at the *in situ* temperature (Figure 5 vs. Figure S1 in Supplemental
information).  In Section 3.4, the relationship between $CO_2$ and DMS gas transfer velocities
and wind speed is examined using $k_{DMS,Sc}$ and $kCO_2$.
## 2.3   Calculation of $k_{bub,CO_2}$
The water-side controlled gas transfer velocity ($k_w$) is comprised of interfacial and bubble-
mediated transfer mechanisms, which operate in parallel, i.e. $k_w = k_{int} + k_{bub}$ (Woolf, 1997).
We assume that turbulence and diffusive mixing at the sea surface operate similarly upon the
interfacial air/sea transfer of $CO_2$ and DMS (i.e. $k_{int,CO2} = k_{int,DMS}$), given appropriate
normalization for the differences in molecular diffusivity.  Observed differences between
$k_{DMS,Sc}$ and $k_{CO_2}$ should therefore be a measure of the difference between the bubble-mediated
contributions to DMS and $CO_2$ gas transfer:
$$\Delta k_w = k_{bub,CO_2} - k_{bub,DMS}$$
*Equation 5*

$k_{bub,CO_2}$ and $k_{bub,DMS}$ are related by the influence of solubility and diffusivity upon bubble-
mediated transfer. We parameterize this relationship simply as $k_{bub,DMS} = f \cdot k_{bub,CO_2}$.
Substitution into Equation 6 yields:
$$k_{bub,CO_2} = \frac{\Delta k_w}{1-f}$$
*Equation 6*



The value of $f$ depends on seawater temperature and the complex dynamics of bubble
formation and cycling (size distributions, surfactants, etc.). At the mean SST encountered in
this study (9.8°C), the Woolf (1997) and Asher et al. (2002) bubble gas transfer models yield
values for $f$ of 0.14 and 0.27, respectively (see Supplemental information for model
equations).
**2.4   Sea surface imaging**
Whitecap areal fraction was measured using images of the sea surface recorded with a digital
camera (5 mega pixel Arecont Vision, 16 mm focal length lens) mounted 14.6 m above the
ocean surface at an angle of ~75° from the nadir. Image footprints represent ~7600 $m^2$ of sea
surface. Images were collected at a sample period of about 1 second and post-processed for
whitecap fraction according to the Automated Whitecap Extraction algorithm method
(Callaghan and White, 2009). Images were further processed to distinguish whitecap pixels
as either stage A or stage B whitecaps by applying a spatial separation technique (Scanlon
and Ward, 2013). The whitecap fraction measurements were averaged in the same way as the
gas transfer velocities (i.e. time-averaged mean values as well as 2 m s$^{-1}$ wind speed bins).
**3   Results**
**3.1   Cruise location and environmental conditions**
This study took place in the summertime North Atlantic (June 24 – July 18, 2011; DOY 175-
199), departing and returning to Woods Hole, MA. Most of the data were collected north of
50°N, including the occupation of four 24-36 hr stations – ST181, ST184, ST187 and ST191
(Figures 1 & 2). The cruise track was designed to sample regions with high biological
productivity and phytoplankton blooms, with large air/sea concentration differences for $CO_2$
and DMS. The cruise meteorology and physical oceanography is discussed in detail by (Bell
et al., 2013). A series of weather systems travelling from West to East passed over the region
during the cruise. Wind speeds ranged from ~1 to 22 m s$^{-1}$, with strongest winds during the
frontal passages at stations ST184 and ST191 (Figure 1b). Atmospheric boundary layer
stability was close to neutral for most of the cruise (|z/L| < 0.07; 75% of the time), with
infrequent stable conditions (z/L > 0.05; <8% of the time). There was no evidence that the
stable periods affected the flux measurements (Bell et al., 2013). Whitecap areal fraction
increased up to a maximum of ~0.06 in response to high wind speeds (Figure 1b).



## 3.2 Whitecaps

Whitecaps were observed during Knorr_11 when wind speeds exceeded 4.5 m s$^{-1}$, a typical wind speed threshold for whitecap formation in the open ocean (Schwendeman and Thomson, 2015; Callaghan et al., 2008). Whitecap areal fraction is a strong, non-linear function of wind speed (Figure 3a). The whitecap vs. wind speed relationship for Knorr_11 is similar in shape, but considerably lower than recent previously published wind speed-based whitecap parameterisations (Schwendeman and Thomson, 2015; Callaghan et al., 2008). At intermediate wind speeds the Knorr_11 whitecap data are as much as an order of magnitude lower than the parameterisations (Figure 3a).

Total whitecap coverage is a function of (i) active 'stage A whitecaps' ($W_A$) produced from recent wave breaking and (ii) maturing 'stage B whitecaps' ($W_B$) that are decaying foam from previous breakers. The Stage A whitecap fraction data is highly variable at ~11 m s$^{-1}$ wind speeds (Figure 3b), which is driven by the difference in the wind-wave conditions during Knorr_11 (ST184 vs ST191, Figure 4a). Stage A whitecap fraction data does not show the same differences between ST184 and ST191 when plotted against the dimensionless Reynolds number, $R_H$, which describes breaking waves using Knorr_11 measurements of significant wave height (Zhao and Toba, 2001). The relationship between Stage A whitecap fraction and $R_H$ is more scattered when Stage A whitecaps are below ~10$^{-4}$ (Figure 4b). Wave development and steepness (slope) influence the likelihood of breaking waves. Breaking waves are more closely associated with steep, young waves. At a given wind speed and wave height, older, swell-dominated waves do not produce as many stage A whitecaps compared to young wave systems (Callaghan et al., 2008; Sugihara et al., 2007).

## 3.3 Concentrations, fluxes and gas transfer velocities

Seawater pCO$_2$ was consistently lower than the overlying atmosphere throughout the study region due to biological uptake (Figure 1c). As a result, the air/sea concentration difference ($\Delta$pCO$_2$) was large and always into the ocean, with $\Delta$pCO$_2$ <-45 ppm for more than 80% of the measurements. Periods with particularly enhanced $\Delta$pCO$_2$ into the ocean were during the transit between ST181 and ST184 ($\Delta$pCO$_2$ as large as -120 ppm) and during ST191 ($\Delta$pCO$_2$ consistently -75 ppm).





Seawater DMS levels were much higher than atmospheric levels, reflecting the biogenic
sources in seawater and the relatively short atmospheric lifetime (~1 day; Kloster et al.,
2006). The largest air/sea DMS concentration differences (ΔDMS) of 6-12 ppb were
observed during DOY 185-190 (Figure 2a). The ΔDMS and ΔpCO$_2$ did not co-vary
(Spearman $\rho$ = 0.11, n=918, $p$<0.001).  This is not surprising because, although seawater
DMS and CO$_2$ signals are both influenced by biological activity, they are controlled by
different processes.  Seawater CO$_2$ levels reflect the net result of community photosynthesis
and respiration, while DMS production is related to metabolic processes that are highly
species-dependent (Stefels et al., 2007).
CO$_2$ fluxes ($F_{CO2}$) were generally into the ocean, as expected given the direction of the air/sea
concentration difference (Figure 1d).  The variability in $F_{CO2}$ observed on this cruise reflects
dependence on both wind speed and ΔpCO$_2$.  For example, during DOY182 air-to-sea CO$_2$
fluxes increase due to a gradual increase in ΔpCO$_2$ with fairly constant wind speed. More
commonly, ΔpCO$_2$ was fairly constant and variability in $F_{CO2}$ reflected changes in wind
speed.  For example, from DOY 185-187 wind speeds gradually declined from ~10 to 5 m s$^{-1}$
with a concurrent decline in $F_{CO2}$. DMS eddy covariance fluxes were always out of the ocean.
Ten minute averaged DMS fluxes ($F_{DMS}$) clearly show the influence of both ΔDMS (e.g.
DOY 188) and wind speed (e.g. DOY 184).
Gas transfer velocities of CO$_2$ and DMS from this cruise exhibit two systematic differences:
i) $k_{DMS}$ values are generally lower than $k_{CO_2}$, particularly during episodes of high wind speed;
and ii) $k_{CO_2}$ is characterized by much larger scatter than $k_{DMS}$. We attribute the large scatter in
$k_{CO_2}$ to the greater random uncertainty associated with the eddy covariance measurement of
air/sea CO$_2$ fluxes compared to those of DMS.  As shown by Miller et al. (2010), the
analytical approach used in this study (dried air, closed path LI7500) has sufficient precision
to adequately resolve the turbulent fluctuations in pCO$_2$ associated with the surface flux over
most of the cruise (ΔpCO$_2$ < -30 ppm).  The scatter in the CO$_2$ flux measurements is more
likely due to environmental variability resulting from fluctuations in boundary layer CO$_2$
mixing ratio arising from horizontal and/or vertical transport unrelated to air/sea flux (Edson
et al., 2008; Blomquist et al., 2014).  These effects likely have a much smaller effect on
air/sea DMS fluxes, because the air/sea DMS concentration difference is always much larger
than the mean atmospheric DMS concentration (due to the short atmospheric lifetime of





DMS). For example, a $\Delta pCO_2$ of 100 ppm at a wind speed of 10 m s$^{-1}$ will produce turbulent
fluctuations that are ~0.02% of the background $CO_2$ on average. In contrast, a typical
seawater DMS concentration (2.6 nM) at 6 m s$^{-1}$ generates fluctuations of 20% of the
background (Table 1; Blomquist et al., 2012). Thus, $F_{CO2}$ measurements are highly sensitive
to small fluctuations in background $CO_2$ and the relative uncertainty is expected to be much
larger than that for $F_{DMS}$.
**3.4   Comparison of $k_{CO_2}$ and $k_{DMS,Sc}$**
The differences between $CO_2$ and DMS gas transfer velocities observed in the time series are
also evident when the data are examined as a function of wind speed. From the 10-minute
averaged data, it is clear that $k_{CO_2}$ is greater than $k_{DMS}$ and has a stronger wind speed-
dependence over most of the wind speed range (Figure 5a,b). These broad trends are also
easily seen in longer time-averaged data. Flux and $\Delta C$ measurements were averaged into 4
hour periods (minimum of 3 flux intervals per 4 hour period), which reduced the scatter in
$F_{CO2}$ while preserving the temporal variability (Figure S3). Gas transfer velocities were then
recalculated from the 4 hour averaged data. 10-minute $k_{CO_2}$ and $k_{DMS,Sc}$ data were also
averaged into 2 m s$^{-1}$ wind speed bins, with a minimum of 5 ten minute periods per bin. The 4
hour averaged data and the wind speed binned data show $k_{CO_2}$ and $k_{DMS,Sc}$ diverging at
intermediate wind speeds, differing by a factor of roughly two at 10 m s$^{-1}$ (Figure 5c,d).
DMS gas transfer velocities on this cruise exhibit complex behaviour at intermediate to high
wind speeds, as discussed in Bell et al. (2013). $k_{DMS,Sc}$ increases linearly with wind speed up
to ~11 m s$^{-1}$ (Figure 5). Under the high wind, high wave conditions encountered during
ST191, the wind speed-dependence of $k_{DMS,Sc}$ was lower than expected, with a slope roughly
half that of the rest of the cruise data. This effect was not observed at ST184. Such coherent
spatial-temporal variation means that wind speed bin averaging of the higher wind speed
$k_{DMS,Sc}$ may mask real variability in the relationship with wind speed. Relationships
developed from wind speed bin-averaged gas transfer data should be interpreted with caution,
especially when it comes to developing generalizable air/sea gas transfer models.
The Knorr_11 $k_{CO_2}$ data also demonstrate a clear wind speed dependence (Figure 5). The
NOAA COARE model for $CO_2$ has been tuned to previous eddy covariance flux





measurements (McGillis et al., 2001), with bubble-mediated transfer determining the non-
linear relationship with wind speed (Fairall et al., 2011). There is reasonable agreement
between the COARE model gas transfer velocity predictions and the Knorr_11 $k_{CO_2}$ data
until ~11 m s$^{-1}$ wind speed. Above 11 m s$^{-1}$, the COARE model over predicts $k_{CO_2}$. This
could be interpreted as indicating high wind speed suppression of gas transfer for $CO_2$ as
observed for DMS (as discussed by Bell et al., 2013).  However, it is important to note that
the number of high wind speed (>15 m s$^{-1}$) gas transfer measurements in this study is limited
to 9 hours and 16 hours of data for DMS and CO2 respectively. Much more data are needed
in order to firmly establish the high wind speed behaviour.
The COAREG 3.1 model parameterizes interfacial gas transfer by scaling to $Sc$ and friction
velocity and estimates bubble-mediated gas transfer following Woolf (1997).  The lower
solubility of $CO_2$ leads to enhanced gas transfer relative to that of DMS at high wind speeds
where bubble transport is significant (Fairall et al., 2011). There is good agreement between
the COAREG model gas transfer velocity predictions and the Knorr_11 $k_{CO_2}$ and $k_{DMS}$ data
until ~11 m s$^{-1}$ wind speed.
Earlier in this paper we introduced the quantity $\Delta k_w$ as an observational measure of the
difference in gas transfer velocities of $CO_2$ and DMS (Section 2.3, equation 6).  The
relationship between $\Delta k_w$ and wind speed is positive and shows no systematic differences
related to temporal variability (Figure 6). Sea surface temperature (SST) is indicated by
symbol size. Some of the scatter in Figure 6 could be driven by changes in $Sc$ due to SST
variability. Nearly all of the data in Figure 6 is from periods when SST was relatively
constant (9.8±1.0°C). Many of the $k_{CO_2}$ data with warm seawater (i.e. ST181, SST > 12°C)
were rejected by our quality control criteria (see Section 6.3). These data were collected when
wind speeds were low, which resulted in small $CO_2$ fluxes with large variability at low
frequencies. Of the periods with SST > 12°C that passed the quality control criteria, the
majority contributed fewer data within a 4 hour averaging period than the minimum threshold
(three 10 minute averaged data points). Only one 4 hour period passed the thresholds for flux
quality control and number of points, and this was associated with the most negative $\Delta k_w$
value.





## 4   Discussion

The bubble-mediated component of gas transfer is a strong function of wind speed and breaking waves. Previous estimates of bubble-mediated air/sea gas exchange have been based on laboratory experiments (Asher et al., 1996; Woolf, 1997; Keeling, 1993). The differences between gas transfer velocities for DMS and $CO_2$ provide a unique way to constrain the importance of bubble-mediated transfer under natural conditions. This study shows that $\Delta k_w$ is near zero at very low wind speeds ($U_{10} \leq 4.5$ m s$^{-1}$), which is consistent with the wind speed at which whitecap fraction becomes significant ($> 10^{-5}$, Figure 3a). Above 4.5 m s$^{-1}$, $\Delta k_w$ increases non-linearly, consistent with an increase in bubble-mediated $CO_2$ transfer associated with wave breaking. The relationship between $\Delta k_w$ and wind speed is non-linear, and the quadratic wind speed-dependence yields a good fit ($R^2 = 0.77$; Figure 6):

$$\Delta k_w = 0.157 U_{10}^2 - 0.535 U_{10} + 4.289 \qquad \textit{Equation 7}$$

The functional form of this relationship is qualitatively consistent with those found between $U_{10}$ and breaking waves/wave energy dissipation (Melville and Matusov, 2002) and $U_{10}$ vs. whitecap areal fraction (e.g. Callaghan et al., 2008; Schwendeman and Thomson, 2015). Bubble-mediated gas transfer is the only viable explanation for the magnitude and wind-speed dependence of $\Delta k_w$. The only alternative explanation would require a large systematic bias in the measurement of relative gas transfer velocities of DMS and $CO_2$. There are no obvious candidates for such biases.

During strong wind/large wave conditions, the Knorr_11 data suggest that bubble-mediated exchange is a dominant contributor to the total transfer of $CO_2$. For example, when wind speeds were 11-12 m s$^{-1}$, $\Delta k_w$ was about 50% of the total $CO_2$ gas transfer ($k_{CO_2}$). A significant contribution by bubbles to the total gas transfer velocity means that bubble-mediated exchange must be included and adequately parameterised by gas transfer models. The Schmidt number ($Sc$) normalisation (Equation 4) assumes that the gas transfer velocity is purely interfacial. An alternative normalisation (involving $Sc$ and solubility) is required when bubble-mediated transfer is significant. Our data suggest that the current $Sc$ normalisation should be applied with caution to gas transfer data for different solubility gases at wind speeds greater than 10 m s$^{-1}$.

If $\Delta k_w$ reflects the difference between the bubble-mediated contribution to the transfer of $CO_2$ and DMS, one would expect $\Delta k_w$ to correlate with wave-breaking, and hence with the areal



coverage of whitecaps. Breaking waves generate plumes of bubbles (Stage A whitecaps,
$W_A$), which then rise to the surface and persist for a short period as foam (Stage B whitecaps,
$W_B$). Almost all whitecap measurements represent the fraction of the sea surface that is
covered by bubble plumes and/or foam i.e. $W_T = W_A + W_B$. $\Delta k_w$ is positively correlated with
both $W_T$ (Spearman $\rho = 0.81$, n=32, $p<0.001$) and $W_A$ (Spearman $\rho = 0.82$, n=26, $p<0.001$)
(Figure 7a,b). These correlations are approximately the same strength as the correlation
between $\Delta k_w$ and wind speed (Spearman $\rho = 0.83$, n=55, $p<0.001$). The functional form of the
relationship between $\Delta k_w$ and whitecap areal extent appears to be linear. However, the
Knorr_11 dataset is small and quite scattered. More data are required to fully test the validity
of whitecap areal fraction as a proxy for bubbles and bubble-mediated exchange.
Observations of the decaying white cap signal ($W_B$) suggest that the persistence of surface
foam is related to sea surface chemistry (Callaghan et al., 2013). $W_B$ is approximately an
order of magnitude larger than $W_A$ and thus dominates the $W_T$ signal. It is often assumed that
gas exchange takes place in bubble plumes formed by active wave breaking (i.e. $W_A$), while
$W_B$ may vary widely due to surfactant concentration with little or no impact upon bubble-
mediated gas exchange (e.g. Pereira et al., 2016). In this case, $\Delta k_w$ should be more strongly
correlated with $W_A$ than $W_B$ or $W_T$. The Knorr_11 data do not suggest that $W_A$ is an
improvement upon either $W_T$ or even wind speed as a measure of bubble mediated exchange.
This may be because whitecaps do not fully represent the bubbles facilitating gas exchange as
these may dissolve before they reach the sea surface. Alternatively, $W_T$ and $W_A$ may be
equally good (or poor) proxies for bubbles because: (i) surfactant activity was minimal in the
study region (despite high biological productivity) such that $W_B$ does not confound the
relationship between $W_T$ and $W_A$; (ii) $W_A$ is no better than $W_T$ at representing the volume of air
entrained by breaking waves; and/or (iii) bubbles residing at the surface (i.e. $W_B$) continue to
contribute to gas transfer (Goddijn-Murphy et al., 2016).
As shown earlier, the bubble-mediated contribution to gas transfer ($k_{bub,CO_2}$) can be obtained
from $\Delta k_w$ using information from mechanistic bubble gas transfer models ($f$, see Section 2.3).
The $k_{bub,CO_2}$ datasets derived from the Knorr_11 data using the Asher et al. (2002) and Woolf
(1997) models differ by about 15% (Figure 8). The field-based estimates of $k_{bub,CO_2}$ can also
be compared to model-only estimates for the Knorr_11 conditions using the Asher et al.
(2002) and Woolf (1997) models. Both models are based on whitecap areal fraction, $W_T$. A




non-linear fit of the Knorr_11 $W_T$ and wind speed measurements ($W_T = 1.9 \times 10^{-6} U_{10n}^{3.36}$) was
used to drive both models (Figure 8). Asher et al. (2002) is based on laboratory tipping
bucket gas evasion experiments (Asher and Wanninkhof, 1998) and the model was then
adjusted to represent the flux of $CO_2$ into the ocean (invasion). Woolf (1997) scaled a single
bubble model to the open ocean based on laboratory experiments.
Both models significantly underestimate $k_{bub,CO_2}$ at wind speeds below about 11 m s$^{-1}$. At
higher wind speeds, the Asher et al. (2002) model increases rapidly with wind speed to agree
slightly better with the Knorr_11 data. In contrast, Woolf (1997) consistently underestimates
$k_{bub,CO_2}$ at all wind speeds. A 'dense plume model' was also developed by Woolf et al. (2007)
to take account of the interaction of a bubble plume with the interstitial water between
bubbles. This model yields estimates of $k_{bub,CO_2}$ that are even lower than the original Woolf
(1997) 'single bubble model' (data not shown).
It is likely that the Knorr_11 cruise data will be compared with estimates of $k_{bub,CO_2}$ derived
from future field campaigns, which will be conducted under different environmental
conditions. Our $k_{bub,CO_2}$ data is at *in situ* seawater temperature (~10°C) and thus *in situ* $CO_2$
solubility ($\alpha$=1.03) and diffusivity ($Sc$=1150). We use the Asher et al. (2002) and Woolf
(1997) bubble models to make estimates of $k_{bub,CO_2}$ normalised to a standard seawater
temperature of 20°C ($k_{bub,CO_2,20°C}$, where $\alpha$=0.78 and $Sc$=666). The 4 hour averaged Knorr_11
cruise data, including estimates of $\Delta k_w$, $k_{bub,CO_2}$ and $k_{bub,CO_2,20°C}$, are provided in Supplemental
Table S1.
The approach used in this study to estimate $\Delta k_w$ and $k_{bub,CO_2}$ from the Knorr_11 field data
neglect the effect of sea surface skin temperature and $CO_2$ chemical enhancement. Skin
temperature is typically only a few tenths of a degree less than bulk seawater under the
conditions encountered in this study (Fairall et al., 1996). The impact upon $k_{CO_2}$ due to skin
temperature effects on $CO_2$ solubility and carbonate speciation is likely on the order of 3%
(Woolf et al., 2016). There is a chemical enhancement of the $CO_2$ flux due to ionization at
the sea surface (Hoover and Berkshire, 1969). The effect on $k_{CO_2}$ has been estimated to be up
to about 8% at a wind speed of 4-6 m s$^{-1}$ (Wanninkhof and Knox, 1996), which amounts to a
maximum impact of a few cm hr$^{-1}$. By neglecting these effects we have slightly





overestimated $\Delta k_w$ and $k_{bub,CO_2}$, but the magnitude of these corrections would be small relative
to the environmental scatter or measurement uncertainty.
**5    Conclusions**
The Knorr_11 concurrent measurements of DMS and $CO_2$ gas transfer velocities show
significant differences in gas transfer between the two gases at intermediate-high wind
speeds.  These data indicate that: i) bubble-mediated gas transfer becomes significant for $CO_2$
at or above the threshold for wave-breaking; and ii) the wind speed-dependence is non-linear,
with a similar functional form to proposed relationships predicting whitecap areal extent from
wind speed. However, existing models of bubble-mediated gas transfer using the Knorr_11 *in*
*situ* observations of whitecap fraction significantly underestimate the importance of this
process.
There are a number of assumptions behind model estimates of bubble-mediated gas exchange
(Goddijn-Murphy et al., 2016). Model bias can be crudely split into: i) uncertainties in the
scaling of whitecap fraction to the bubble population (e.g. using Cipriano and Blanchard,
1981); and ii) the relationship between gas exchange and bubble properties, which are
predicted as a function of air entrainment into the surface ocean by a breaking wave, bubble
injection depth, size distribution and mobility through the water (a function of surface
cleanliness and surfactants). The underestimation of bubble-mediated $CO_2$ gas transfer by
both models is particularly apparent at low-intermediate wind speeds and low whitecap
fraction. This could indicate that either bubble production during microscale breaking is an
important process for gas transfer or the relationship between whitecap fraction and the
bubble population is poorly constrained.
In summary, the approach of using simultaneous measurements of multiple gases with
different solubility appears to be a viable way to constrain the magnitude of bubble-mediated
gas transfer.  Analysis of additional sparingly soluble gases, such as methane or oxygenated
hydrocarbons would further strengthen this approach.  A much larger data set, under a wider
range of oceanographic conditions is certainly needed.  In particular, it would be useful to
examine DMS and $CO_2$ gas transfer velocities in ocean regions with different temperatures,
where the solubility of each gas is significantly different from this study.





**Appendix A**

**A.1  Seawater CO$_2$ and DMS measurements**

Seawater CO$_2$ and DMS were monitored in the supply of seawater pumped continuously through the ship from an intake on the bow located 6 m below the sea surface. CO$_2$ was equilibrated with air in a recirculating showerhead-type system. Alternate air and water side pCO$_2$ were each measured for 5 min by the same Infrared Gas Analyser (IRGA). Seawater DMS was equilibrated with DMS-free air in a tubular porous membrane equilibrator, operated in a single-pass, counterflow mode.  DMS was measured at 1 Hz using chemical ionization mass spectrometry and bin-averaged at 1 minute intervals (UCI miniCIMS; Saltzman et al., 2009).  DMS was calibrated by continuously pumping an internal standard of tri-deuterated, DMS (d3-DMS) into the seawater flow just before the equilibrator.  Details of the methods and instrumentation used for equilibration and detection of seawater DMS are described in Saltzman et al. (2009).

**A.2  Mast-mounted instrumentation and data acquisition**

The eddy covariance setup was mounted 13.6 m above the sea surface on the bow mast. Platform angular rates and accelerations were measured by two Systron Donner Motion Pak II (MPII) units.  Three dimensional winds and sonic temperature were measured by two Campbell CSAT3 sonic anemometers.  Air sampling inlets for DMS and CO$_2$ were located at the same height as the anemometers and within 20 cm of the measurement region.  GPS and digital compass output were digitally logged at 1 Hz. Winds were corrected for ship motion and orientation as described in Miller et al. (2008) and Landwehr et al. (2015).  The eddy covariance data streams were logged in both analog and digital format as described in Bell et al. (2013) and Miller et al., In Prep.

**A.3  High frequency atmospheric DMS and CO$_2$ measurements**

Atmospheric DMS measurements were made at 10 Hz using an atmospheric pressure chemical ionisation mass spectrometer located in a lab van (UCI mesoCIMS; Bell et al. (2013)).  Air was drawn to the instrument through a 28 m long ½ in OD Teflon tube.  A subsample of the air stream was passed through a Nafion drier prior to entering the mass



spectrometer. The measurement was calibrated using an internal gas standard of tri-
deuterated DMS added to the inlet (see Bell et al., 2013).
Atmospheric $CO_2$ measurements were made on air drawn at 8 L min$^{-1}$ through a filtered inlet
(90 mm diameter with 1 micron pore size, Savillex) near the sonic anemometers on the bow
mast, through 5 m of 5.9 mm ID polyethylene-lined Dekabon tubing to two fast-response
$CO_2/H_2O$ IRGAs in an enclosure on the bow mast. The IRGAs were open-path style sensors
(LI7500, Licor Inc.) converted to a closed-path configuration (see Miller et al., 2010) and
were plumbed in series. A Nafion multi-tube membrane drier (PD-200T, PermaPure) with 6
L min$^{-1}$ dry air counter flow was installed between the two IRGAs such that the upstream
IRGA sampled undried air and the downstream IRGA sampled the same air after drying. This
technique removes 97% of the Webb Correction from the measured $CO_2$ flux (first shown by
Miller et al. (2010) and confirmed by Landwehr et al. (2014)).
The air flow through both the $CO_2$ and DMS inlets was fully turbulent (Re > 10,000). The
inlets used in this study introduced a small delay ($\Delta t = 2.2$ s for DMS, $\Delta t = 1.2$ s for $CO_2$)
between measured wind and atmospheric measurements, as well as minor loss of covariance
at high frequencies (<5%). The methods used to estimate the delay and loss of flux are given
in Bell et al. (2013).
Eddy covariance fluxes were computed for DMS and $CO_2$ as $F_{DMS}$ or $F_{CO2} = \sigma_{air} \langle w'c' \rangle$
where $\sigma_{air}$ is the dry air density, $w'$ is the fluctuation in vertical winds and $c'$ is the delay-
adjusted fluctuation in gas concentration. Average covariance fluxes were processed in 10
minute and 9.5 minute intervals for DMS and $CO_2$, respectively (hereafter referred to as 10
minute intervals). Momentum and sensible heat fluxes were also computed for 10 minute
intervals (see Bell et al., 2013).
Sampling intervals with a mean wind direction relative to the bow of >90° were excluded
from the final data set. $CO_2$ fluxes were also excluded from intervals when either: i) relative
wind direction changed excessively (SD > 10°); ii) relative wind speed was low (< 1 m s$^{-1}$);
or iii) $\Delta CO_2$ was low (< |30| ppm). DMS and $CO_2$ fluxes were quality controlled for excessive
low frequency flux as described in the Supplemental information of Bell et al. (2013). These
quality control criteria excluded 62% of the intervals for $CO_2$ and 55% for DMS and
significantly reduced the scatter in the data.





*Acknowledgements.* We thank the Captain and crew of the R/V Knorr and the Woods Hole
Marine Department for their assistance in carrying out this cruise. Funding for this research
was provided by the NSF Atmospheric Chemistry Program (AGS-0851068, -0851472, -
0851407 and -1134709) and the NSF Independent Research and Development program.
B.W. acknowledges support from Science Foundation Ireland under grant 08/US/I1455 and
from the FP7 Marie Curie Reintegration programme under grant 224776. This study is a
contribution to the Surface Ocean Lower Atmosphere Study (SOLAS).

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





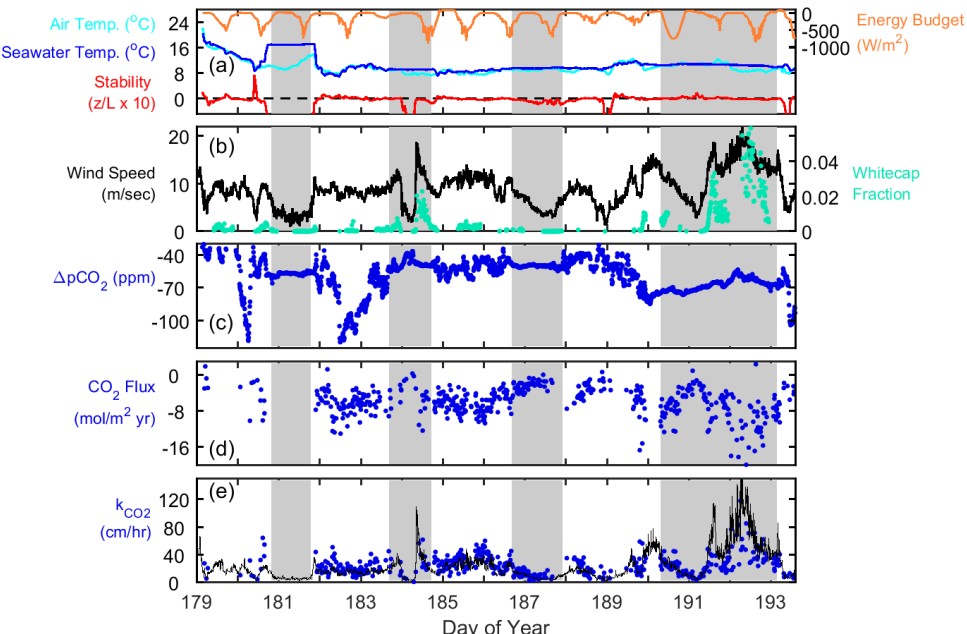


**Figure 1:** Time series of ten minute averaged data collected during the Knorr_11 cruise. Dashed
black line in panel (a) indicates neutral atmospheric stability (z/L = 0). Grey shaded regions represent
intervals when the ship occupied stations ST181, ST184, ST187, and ST191. Panels (c), (d) and (e)
are the $CO_2$ concentration difference ($\Delta pCO_2$), flux ($F_{CO2}$) and gas transfer velocity ($k_{CO2}$) (water-side
only, no $Sc$ correction), respectively. Panel (e) also shows $k_{CO2}$ calculated using the NOAA COARE
model (black line). Note that negative $k_{CO2}$ data points in (e) were omitted for clarity (see
Supplemental Figure S2 for full data set).

665





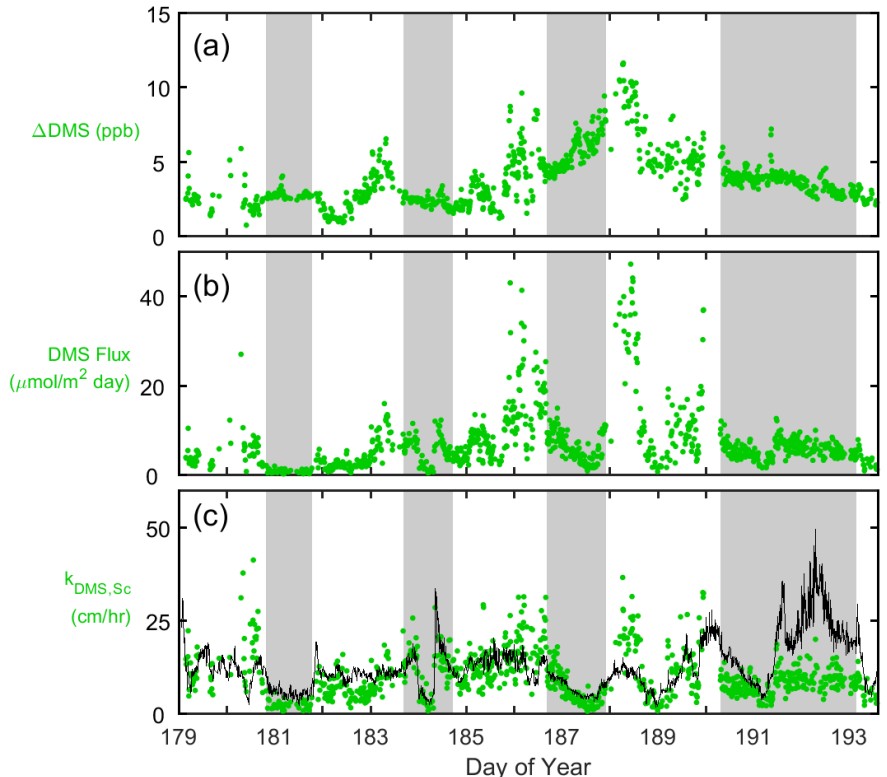

666

**Figure 2:**  Knorr_11 cruise time series of ten minute averaged DMS: (a) air/sea concentration difference ($\Delta$DMS); (b) flux ($F_{DMS}$); and (c) gas transfer velocity normalised to the *in situ* $CO_2$ *Sc* number ($k_{DMS,Sc}$). Panel (c), shows $k_{DMS,Sc}$ calculated using NOAA COARE model output (black line). Grey shaded regions represent periods on station.

671





672

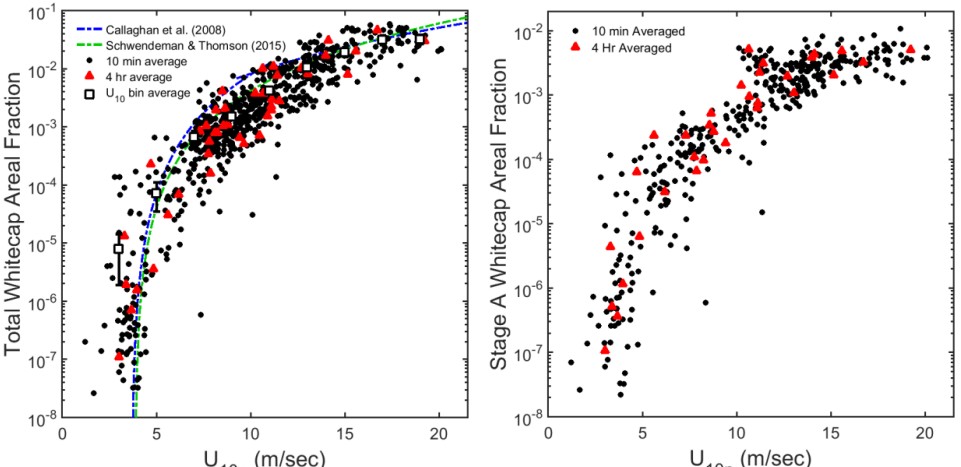

673

**Figure 3:** Semi-log plots of whitecap areal fraction as a function of mean horizontal wind speed at 10 m above the sea surface ($U_{10}$) during the Knorr_11 cruise. 10 min average (black dots) and 4 hour average (red triangles) data are shown on both panels. Left panel shows total whitecap area versus $U_{10}$ bin averaged data (open squares, 2 m s$^{-1}$ bins). Wind speed parameterisations from the recent literature are shown for reference. Right panel is the whitecap area considered to be solely from wave breaking (Stage A whitecaps, see text for definition).

680





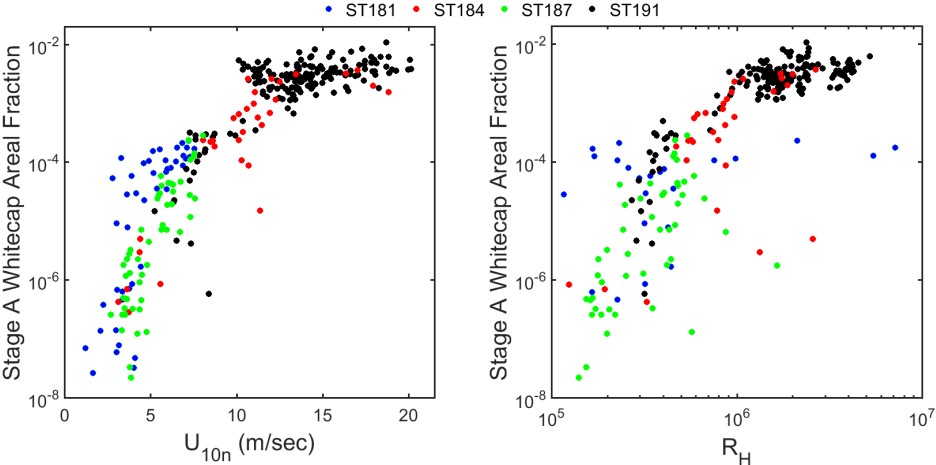

681

**Figure 4:** Semi-log plots of Stage A whitecap areal fraction as a function of wind speed ($U_{10}$, left panel) and as a function of a non-dimensional Reynolds breaking wave parameter $R_H$ (right panel), calculated from Knorr_11 measurements of significant wave height (Zhao and Toba, 2001). Plots show data only from when the ship was on station, segregated into ST181 (blue), ST184 (red), ST187 (green) and ST191 (black). The highly variable Stage A whitecap fraction vs. $U_{10}$ at ~11 m s$^{-1}$ is driven by differences in the wave environment during ST184 and ST191. Stage A whitecap fraction vs. $R_H$ exhibits no bimodal behaviour and there is no clear difference between ST184 and ST191. The relationship between Stage A whitecap fraction and $R_H$ is more scattered when Stage A whitecaps are below ~10$^{-4}$.


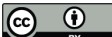



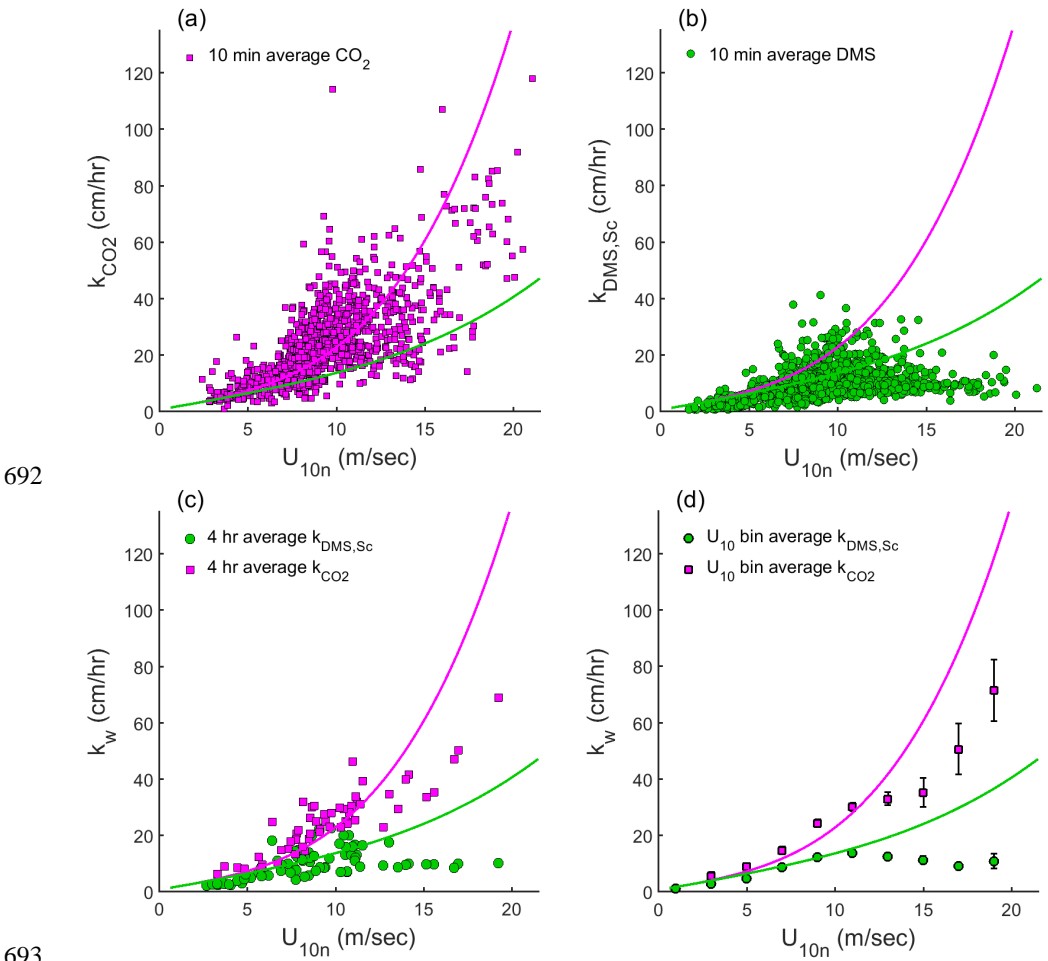




**Figure 5:** Gas transfer velocities plotted against mean horizontal wind speed ($U_{10}$) from the Knorr_11 cruise. Ten minute average data for $CO_2$ (a) and DMS (b). DMS gas transfer velocities are normalised to the *in situ* $CO_2$ *Sc* number. Data are averaged into 4 hour periods (c) and 2 m s$^{-1}$ wind speed bins (d). Note that negative $k_{CO2}$ data in (a) and (c) have not been plotted for clarity (see Supplemental Figure S4 for full data set). For reference, the NOAA COAREG3.1 model output for $CO_2$ (magenta line) and DMS (green line) is plotted on all four panels. The COARE model was run with the turbulent/molecular coefficient, A = 1.6, and the bubble-mediated coefficient, B = 1.8, and used mean Knorr_11 data for the input parameters.


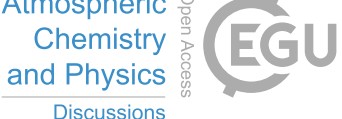



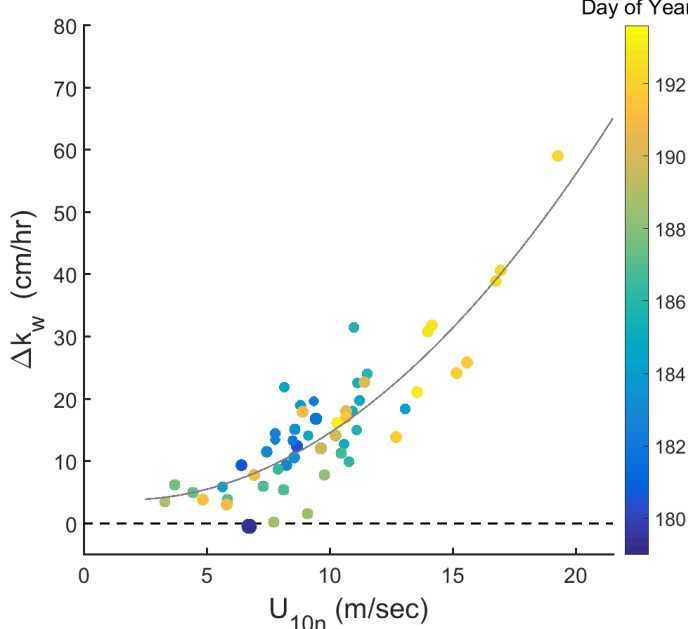


**Figure 6:** Difference ($\Delta k_w$) between 4 hour average $k_{CO_2}$ and $k_{DMS,Sc}$ plotted against $U_{10}$. Data are
coloured by the date of measurement (Day of Year). The solid grey line describes a cubic fit to the
data (see text for coefficients).


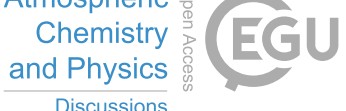

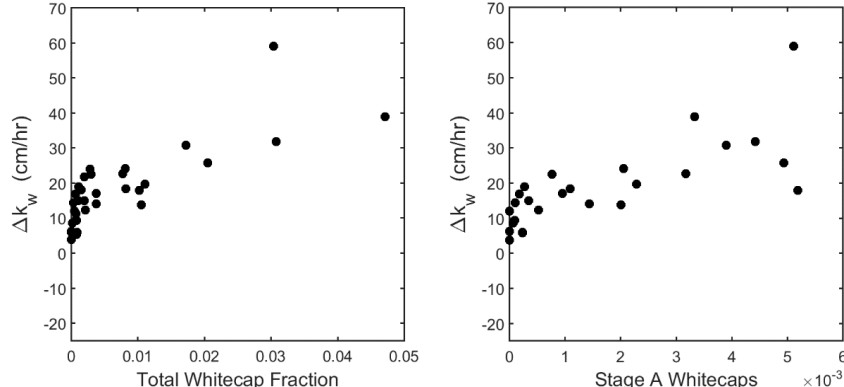


**Figure 7:** Knorr_11 $\Delta k_w$ data plotted against total whitecap areal fraction (left panel) and against Stage
A whitecap areal fraction (right panel). Each point is a 4 hour average of coincident measurements of
whitecap fraction and DMS and $CO_2$ gas transfer.






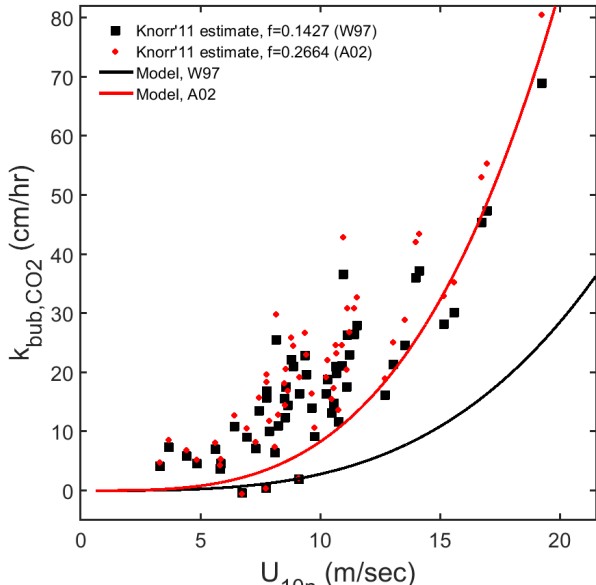


**Figure 8:** Bubble-mediated transfer velocity of $CO_2$ ($k_{bub,CO_2}$) as a function of wind speed.
Individual points are Knorr_11 observations using solubility and diffusivity scaling from Woolf
(1997) (black squares) and Asher et al. (2002) (red circles). Continuous lines are model calculations
of $k_{bub,CO_2}$ using the Knorr_11 wind speed-whitecap areal fraction relationship and mean SST (Woolf
(1997), black; Asher et al. (2002), red).

720