# Peer review of "Estimation of bubbled-mediated air/sea gas exchange from concurrent DMS and CO2 transfer velocities at intermediate-high wind speeds"

_Atmospheric Chemistry and Physics, 2017_

## Short Comment (SC1) · 14 Feb 2017

I'm glad to see this analysis. We've been puzzling over significant differences between Knorr11 results and the 2013 HiWinGS cruise (also on the Knorr). The discussion is helpful.

One comment on section 2.2. I believe it's incorrect that air-side resistance is a function of gas solubility. It depends only on diffusivity in air (or Sc_air), and is about the same for all gases. The physics of air-side mass transfer are fairly well understood, and you are using COAREG to estimate ka for DMS. You should find computed ka (or ra) for

$CO_2$ is almost identical to ka for DMS, and it could simply be subtracted from K to get kw. Although the correction is a much smaller fraction of Kco2.

The striking feature of these measurements are the low values for both kdms and kco2 at ST191 (the majority of high wind conditions sampled on this cruise). Compared to the 2013 HiWinGS observations, k660 for both DMS and $CO_2$ on Knorr11 are ∼30 cm/hr lower in the highest wind speed bin (19 +/- 1 m/s). Sea state conditions conditions during ST191 may have contributed to a suppression of interfacial transfer for both gases via reduced tangential stress in the presence of large waves, as suggested by prior theoretical papers and wave tank studies. But the proper metric to quantify that effect remains elusive (to me). During HiWinGS we saw no suppression in k for either gas in wind speeds up to 25 m/s and wave heights up to 8m, and no obvious trends with wave age. So I'm wondering what significant difference existed between the two cruises? You might consider adding sea state parameters to Table S1 (Hs, Cp, wave age, whitecap fractions, etc.).

You might examine at how COAREG is computing whitecap fraction for the kb calculation. From the plotted curves, it looks like Wf is an ∼cubic function of wind speed. You could try replacing that with a whitecap model based on more recent measurements, or with an empirical fit to the Knorr11 observed Wf. You could also simply use your measured Wf in the computation of kb. Any of that would require retuning the B parameter for a best fit to both gases, especially if you use stage A Wf.

However, current versions of COAREG don't consider reduction in tangential stress with flow separation (related to, for example, wave age), so assuming flow separation is causing suppression of k at high winds, it's unlikely to cleanly fit this data set . . .

Last, it would be nice if the plots had gridlines. . .

Cheers, B. Blomquist

---

## Referee Comment (RC1) · I. Brooks (Referee) · 27 Feb 2017

Review of: ACP-2017-85 - "**Estimation of bubbled mediated air/sea gas exchange from concurrent DMS and CO2 transfer velocities at intermediate-high wind speeds" by T. G. Bell et al.**

Reviewed by Ian Brooks

This paper presents direct measurements of the air-sea fluxes and transfer velocities of both $CO_2$ and DMS from a cruise in the North Atlantic. The difference in behaviour of the transfer velocities as functions of wind speed has been widely assumed to be a result of the different contributions of bubble mediated fluxes for gases of very different solubility. Here, simultaneous measurement of both gas fluxes allows a direct estimate of the bubble mediated flux to be calculated for open ocean conditions, and a comparison to model predictions to be made.

This is an important contribution to the field, and deserving of publication after revision for, mostly minor, deficiencies detailed below.

DETAILED COMMENTS:

Line 86-87: "In that study, no statistically significant difference was observed in gas transfer-wind speed relationships of CO2 and DMS for winds below 10 m s$^{-1}$" – need to clarify this statement, was a significant difference found for winds above 10 m s$^{-1}$? Was 10 m s$^{-1}$ the maximum wind speed in the study?

Line 104-105: as noted in the comment from Blomquist, the air-side resistance is not a function of solubility (though its contribution to the derivation of waterside transfer velocity is dependent on solubility).

Line 122-123: The use of the COAREG 3.1 model to calculate air-side transfer velocities in order to derive the waterside transfer velocity introduces an assumption that COAREG is providing valid values of $k_a$. Any uncertainty in this will impact the later results and should be acknowledged and if possible quantified.

Line 126-127: "The air side gas transfer contributes about 5% on average to the total resistance for DMS" – do you mean 'air-side resistance' rather than air-side transfer?

Line 140: "...(Equation 4)..." should be "...(Equation 3)..."

Line 170: "...into Equation 6 yields..." should be "...into Equation 5 yields..."

Line 192 - figures: Figures 1 and 2 are introduced here, but figure 2 is not actually discussed until after discussion of figure 4 breaking the flow of discussion and figures, and leaving me, initially, confused as to how I'd missed the discussion of figure 2. In fact, only the general environmental conditions shown in figure 1 are discussed here, not the gas flux results, which are discussed much later. Figure 1 would be better split into 2, separating the gas fluxes into a figure matching the format of current figure 2. The figures showing the gas flux results could then be placed in a logical order within the discussion. Since wave state is a relevant parameter in the later discussion, it would be useful to add a time series of at least significant wave height to figure 1.

Line 207: The authors note that their estimates of whitecap fraction as a function of wind speed are substantially lower than other recently published values – at times an order of magnitude lower. A likely reason for this is the exposure settings on the camera. During the HiWinGS project cruise in 2013 two independent sets of cameras were used for whitecap imaging. They were initially found to give whitecap fractions that differed by a factor of several. Tests were conducted during the final transit of the cruise, in which a pair of identical cameras were run side by side; one with fixed exposure settings, the other having the exposure settings changed every few hours. The exposure settings were found to make a substantial difference to the whitecap fraction calculated

using the same Callaghan and White (2009) algorithm used here – up to a factor of 4 for the range of settings tested. It was found that almost all of this difference (both between the 2 cameras in the exposure trial, and between the two sets of cameras used throughout the cruise) was removed if the images were first 'normalised' to remove any brightness gradient across the image. Brief details of these tests will be given in

Brumer, S. E., C. J. Zappa, **I. M. Brooks**, H. Tamura, S. M. Brown, B. Blomquist, C. W. Fairall, A. Cifuentes-Lorenzen, 2017: Whitecap coverage dependence on wind and wave statistics as observed during SO GasEx and HiWinGS, *J. Phys. Oceanogr.* (under revision)

Line 213-215: "Stage A whitecap fraction data is highly variable at ~11 m s-1 213 wind speeds (Figure 3b), which is driven by the difference in the wind-wave conditions during Knorr_11 (ST184 vs ST191, Figure 4a)" – two points:
(1) the difference is ascribed to different wind-wave conditions at the two stations, but no wave data are shown. As noted above, relevant wave parameters need to be added to figure 1.
(2) A similar broad range of stage A whitecaps is evident at around $U = 6$ m s$^{-1}$, also resulting from grouping of high/low values by different stations...are the wind-wave conditions similarly different in this case?

Lines 218-223: The authors first note that where stage A whitecap fractions is $< 10^{-4}$ the relationship with $R_H$ is more scattered than at higher fractions; they then note a number of factors that affect wave breaking and so whitecap fraction, but don't make a coherent link back to their initial point about the scatter in the stage-A whitecap / $R_H$ relationship. This reads as an almost unconnected series of statements...all true, but leaving the reader wondering what the point being made is.

Line 282: "...(Figure 5)" -> "...(Figure 5b,d)" – again text refers to high wave conditions for ST19 but no wave data provided for reader to assess.

Figure 5 and the discussion of it have some general issues:
- It's hard to see the pink/green lines against the mass of pink/green dots on panels a and b – it would help here to plot the dots in a paler shade of pink/green to allow the lines to stand out.
- The curves shown, for the COAREG3.1 model are a useful reference, but fits to the actual data are also needed; these would allow a much clearer assessment of how closely the COARE model agrees with the observations.
- lines 282-284: "Under the high wind, high wave conditions encountered during ST191, the wind speed-dependence of $k_{DMS,Sc}$ was lower than expected, with a slope roughly half that of the rest of the cruise data. This effect was not observed at ST184." – since the ST191 data are not highlighted in any way it is not possible for the reader to judge the behaviour here. Of note perhaps is not simply the high wind and wave conditions during ST191 but the different time history of the winds – a sustained period of high winds during ST191 vs a very short period in ST184 where the wind rises rapidly, spikes, and decreases rapidly. These two periods are likely to produce very different wave fields at the same wind speeds – again, reason to plot the wave parameters in figure 1 – which might explain the very different whitecap fractions seen in Figure 1b for these periods.

Line 294 & 305: the phrasing "until 11 m s$^{-1}$ wind speed" is rather clumsy; 'until' implies a variation over time, which is not what is meant – "...up to wind speeds of 11 m s$^{-1}$" would read better.

Line 326: "...$\Delta kw$ is near zero at very low wind speeds ($U10 \leq 4.5$ m s$^{-1}$)..." – this is hard to judge. Eyeballing the data points I would agree; however, there are only 3 points at $U < 4.5$ m s$^{-1}$, and all of those at $U>3$ m s$^{-1}$. Their mean $\Delta k$ is ~5 cm/hr and the fitted curve approaches a $\Delta k$ of ~3 cm/hr at $U = 0$, not zero. One might argue for an alternative functional form in which the exponents were not prescribed might better represent the data; the quadratic used here implicitly assumes the functional dependence. (Note also, the figure caption states that the plotted curve is cubic not quadratic).

What is the reasoning behind using 4-hour averages of transfer velocities here (and elsewhere)? 4 hours is quite a long time relative to the time period over which significant changes in forcing can take place. Granted it greatly reduces scatter, but I would wary of averaging over periods much long than ~1 hour.

Line 358: "...the relationship between $\Delta kw$ and whitecap areal extent appears to be linear." – I'm not convinced this is entirely true – approximately so over the range $0.005 < W < 0.05$, but ~half the data points lie at $W < 0.005$, and seem to drop off rather more rapidly than the fit to the high values of W would indicate. Plotting W on a log scale might give a rather different impression. Not that IF the relationship is linear as suggested then eyeballing a fit (if you claim a linear fit, it would help to show it!) suggests $\Delta k$ = 10-15 cm/hr at W = 0, which raises questions as to why that should be when there are no bubbles to account for a difference in k, and why this minimum difference is several times higher than that derived as a function of wind speed. If on the other hand a roughly linear fit of $\Delta k$ to log(W) existed (which I think the rapid drop off in $\Delta k$ at very low W might support) then $\Delta k$ would approach zero at low W.

Line 366-367: "In this case, $\Delta kw$ should be more strongly correlated with $WA$ than $WB$ or $WT$." – in a general sense, this is true, but is only if the various factors affecting foam decay vary. If foam decay rate is constant then $W_T$ should be proportional to $W_A$.

Line 387: "Both models significantly underestimate $k_{bub,CO2}$ at wind speeds below about 11 m s$^{-1}$." – Actually both models rather underestimate the observationally derived $K_{bub,CO2}$ at all windspeeds; however, note that both models are driven by the observed wind-whitecap relationship, which has already been stated to be low compared to other recent estimates. Is the agreement better using a whitecap function that agrees more closely with the recent consensus?
While the Asher et al. model is lower than the observations, it is not wildly so (essentially matching the lower boundary of the observed values), and the agreement in both values and functional behaviour is rather convincing.

Line 455: "...eddy covariance setup..." -> "...eddy covariance system..."

Line 463: reference to paper in preparation not generally allowed...if it's not 'in press' by the time this manuscript it copy edited, the copy editor will want the reference cut.

Figure 3: it's hard to pick out the curves and black square points against the mass of black dots. Suggest changing black dots to mid-grey and ensuring everything else is plotted over them. It would also be good to see a functional fit to this data as well as the functions from previous studies...especially as this function is later used to drive the $k_{bub,co2}$ models plotted in figure 8.

---

## Short Comment (SC2) · 27 Feb 2017

Byron, many thanks for your comments. Very helpful and some good suggestions. Three responses to your comments:

1. You are correct that air-side resistance is not a function of gas solubility. We will change the sentence on Line 104-105 to: "The relative contribution of air-side resistance to the total resistance is a function of solubility and thus different for the two gases."

2. I agree that the difference between the Knorr11 and HiWinGS measurements could

be driven by differences in the sea state. I will add Hs, Cp, wave age and whitecap fraction (Wf) data to Table S1.

3. I have already plotted kBubble estimates using an empirical fit to the Knorr11 whitecap data (red and black lines in Figure 8). The differences between the Knorr11 wind speed-Wf fit and other fits that use more recent Wf measurements (e.g. Callaghan et al., 2008; Schwendeman and Thomson, 2015) are small. I agree that eventually the COAREG B parameter should be tuned to fit observations. Once more data has been collected I think that this is a good idea. However, given the large discrepancy between the models and these first observations of kBubble, I do not see the value of adjusting the COARE model at this point.

4. I will consider adding gridlines to the plots where possible. However, I will not add gridlines if they make it difficult to see the data on the plots.

References:

Callaghan, A. H., de Leeuw, G., Cohen, L., and O'Dowd, C. D.: Relationship of oceanic whitecap coverage to wind speed and wind history, Geophysical Research Letters, 35, L23609, 10.1029/2008gl036165, 2008. Schwendeman, M., and Thomson, J.: Observations of whitecap coverage and the relation to wind stress, wave slope, and turbulent dissipation, Journal of Geophysical Research: Oceans, 120, 8346-8363, 10.1002/2015jc011196, 2015.
* * *

---

## Referee Comment (RC2) · W. E. Asher (Referee) · 5 Apr 2017

This paper discusses an interesting dataset of field gas transfer experiments where the air-sea fluxes of CO2 and DMS were measured using direct-covariance methods. The authors reduce the data to get at the bubble-mediated fraction of the total gas transfer velocity by assuming that differences in Schmidt-number normalized total gas transfer velocities for the two gases will give the Schmidt-number normalized bubble gas transfer velocities. This relation is given in Equation 5. Then, it is proposed that the two bubble transfer velocities can be scaled using two different relationships for

bubble gas transfer (one by myself and co-workers, and one by D. Woolf).

The approach is novel, but I think the authors are glossing over a potential problem in that in the system they are studying, CO2 is an invasive flux (air-to-ocean) and DMS is an evasive flux (ocean-to-air). My hunch is that Equation 5 is only strictly true when both gases are far from equilibrium *and* the flux is in the same direction. Problems arise in applying Equation 5 for a mixed system, where one gas is invading and one is evading, because the bubble gas flux is not the same, even when normalized to a common diffusivity/solubility. In the case of invasion, the bubble overpressure drives more gas than expected (based on the bulk air-ocean concentration difference) into the water. For evasion however, the bubble overpressure acts to decrease the net gas flux.

This means, at least for my parameterizations, that the functionality of the relationships that determine the dependence on solubility (which is the main difference between the transfer velocity for bubble-mediated processes and transfer across a wavy, unbroken surface) is not the same for invasion and evasion (see Asher et al., 1996, JGR-Oceans). Woolf gets around this issue by defining a diffusivity/solubility-dependent equilibrium supersaturation, which will not be the same for DMS and CO2, and should be taken into account (I think) when applying Equation 5.

It isn't clear to me at this point whether or not Equation 5 is incorrect, or just needs to be qualified that it only holds in the specific case when the two bulk air-sea concentration differences (for CO2 and DMS) are far from equilibrium. However, one thing is clear from looking at the material in the supplements, is that using the Asher et al. (2002) relationship for both CO2(invasion) and DMS (evasion) is not correct. The Asher et al. (2002) relation is only for invasion. For evasion, there is a separate equation in Asher and Wannninkhof (1998, "The effect of bubble-mediated gas transfer on purposeful dual gaseous-tracer experiments." Journal of Geophysical Research 103(C5): 10,555-510,560) that should be used instead. However, I think the authors need to consider whether or not their approach might be flawed from the outset due to the mismatch in flux directions.

Other than a few minor comments and technical details I've listed below, the paper is good and I think the results are interesting. My main comment above also should not be seen as a fatal flaw. Provided the authors can justify Equation 5, and their derivation of f (equation 6), I think this paper could be published with relatively few changes.

Minor comments: Line 54: "These processes include ..." Comment: Buoyancy effects are not a process. It might be better in this sentence to say something like "These processes include diffusion, surface renewal, and bubble-mediated transport. In turn, turbulence can be generated by wind stress, wave-induced mixing, buoyancy currents, and wave breaking." Or something like that anyway.

Line 56: "A variety of theoretical, laboratory, and field ..." Comment: I don't think this sentence is strictly true. My opinion is we have a fairly good understanding of the factors that affect gas exchange from a phenomenological standpoint (the authors list them just a couple of sentences earlier). What we lack is how to determine which of those processes are important under a given set of circumstances. Most of this comes from the fact it is challenging to measure the things we know affect gas exchange in the field, at least at the scales over which these things control gas transfer.

Line 60: "Gas transfer via bubbles (k_bub) ..." Comment: It would be good to define k_bub here. The point is that there are a couple of different ways to do this, you can go the Memery and Merlivat (Memery, L. and L. Merlivat (1985). "Modeling of the gas flux through bubbles at the air-water interface." Tellus, Ser. B 37: 272-285) approach and use the bulk air-water concentration difference and accept that k_bub for invasion and evasion are different (e.g., I used this approach in Asher et al. (1996, JGR-Oceans)) or you can redefine the air-water concentration difference in terms of how bubbles would affect the equilibrium and have a common k_bub (but then it might get complicated relating k_bub for invasion and evasion) as done by Woolf (1997).

Line 78: "These measurements typically show DMS gas transfer velocities that are lower and exhibit more linear wind speed dependence than those estimated for CO2

based on dual tracer studies (e.g. Bell et al., 2015; Yang et al., 2011; Goddijn-Murphy et al., 2012)." Comment: I think the authors should be clear here that there are no CO2 measurements from dual-tracer studies. There are DT measurements for SF6/He, which get related to CO2 through diffusivity. Then there are EC measurements for CO2. Comparison of the DT-derived CO2 transfer velocities with CO2 transfer velocities produced by EC measurements of CO2 fluxes shows relatively good agreement. It is the transfer velocities produced by EC measurements of DMS fluxes that show different behavior.

Line 87: Comment: maybe want to note that they agree when normalized to a common diffusivity.

Line 126: "The air side gas transfer 127 contributes about 5% on average to the total resistance for DMS." Comment: The air-side resistance fraction is a function of wind speed. Does this 5% increase as U increases? COAREG must reproduce this, it was measured by McGillis et al. a while back.

Line 161: the relation in the text showing k_w = k_int + k_bub. Comment: I wonder if maybe it is time to stop writing this as a general expression (I know, I am guilty of this as well). What is generally true is that the total gas flux is equal to the sum of interfacial flux and the bubble flux. Saying the overall transfer velocity is equal to the sum of the two transfer velocities really only works if the concentration difference is far from equilibrium. Work through David's relations from the 1997 paper and you'll find they are a bit convoluted in terms of how exactly the pieced (his Delta term) fit together to make a coherent physical picture. If you start by assuming it is the fluxes, not the transfer velocities, which sum linearly, the assumptions required to get to the various relations proposed are more easily understood.

Technical Comments:

1. Multiple citations are not in any recognizable order. Sometimes they are chronological, sometimes alphabetical. I don't remember what the ACP style guide says, but I

am sure it is not "random."

2. Line 74: "... studies indicate a non-linear dependence ..."

Line 91: Shouldn't cite papers that are not published or submitted.

Line 175: The two f values are opposite from what is given in the supplement. Not sure which is correct, but it should be consistent (and correct).

Line 323: I know this is petty, but I don't think Woolf (1997) is based on laboratory data.

Equation 7: Figure caption says "cubic" and equation 7 is quadratic. Resolve this difference.

Line 384: The citation to Asher and Wanninkhof (1998) should be to Asher et al. (1996). If you really must cite Asher and Wanninkhof (1998) in this context, which you shouldn't, at least make it the other Asher and Wanninkhof (1998) paper that is directly relevant (see citation above).

---

## Author Comment (AC1) · 6 Jun 2017

We thank both reviewers for their positive and constructive comments. We address their specific points below:

**Reviewer 1: Bill Asher**

*I think the authors are glossing over a potential problem in that in the system they are studying, $CO_2$ is an invasive flux (air-to-ocean) and DMS is an evasive flux (ocean-to-air). My hunch is that Equation 5 is only strictly true when both gases are far from equilibrium \*and\* the flux is in the same direction. Problems arise in applying Equation 5 for a mixed system, where one gas is invading and one is evading, because the bubble gas flux is not the same, even when normalized to a common diffusivity/solubility. In the case of invasion, the bubble overpressure drives more gas than expected (based on the bulk air-ocean concentration difference) into the water.*

This is true, and we are glad that you brought it up. However, it is a small effect. Woolf (1997) provides the means to calculate the magnitude of this effect. The average fractional extra pressure on the gas in contact with the sea ($\Delta$) can be estimated from:

$$\Delta = (U/U_i)^2 \%$$

where Ui is the wind speed at which the supersatuation of a particular gas equals 1% (49 m s$^{-1}$ for $CO_2$). A high wind speed (20 m s$^{-1}$) gives $\Delta = 0.167\%$. An atmospheric $pCO_2$ of 400 ppm implies that the bubble overpressure would be 0.67 ppm. This is a ~2% enhancement of the $CO_2$ flux when the air/sea concentration gradient is small (minimum for this study = 30 ppm). Larger air/sea concentration gradients would diminish the magnitude of the bubble overpressure further.

We have added the following text to our revised manuscript after equation 5:
"Strictly speaking, Equation 5 should also account for the influence of bubble overpressure, which alters the gas flux due to bubbles when the concentration gradient is toward the ocean. The extra pressure on the gas in the bubbles is calculated following Woolf (1997): $\Delta = (U_{10}/U_i)^2$ % where $U_i$ is the wind speed at which the supersaturation of a particular gas equals 1% (49 m s$^{-1}$ in the case of $CO_2$). A high wind speed (20 m s$^{-1}$) gives $\Delta = 0.167\%$, which would lead to a ~2% enhancement of the $CO_2$ flux when the air/sea concentration gradient is 30 ppm (minimum for this study) and into the ocean. The magnitude of this effect would be larger for gases less soluble than $CO_2$ but we are able to ignore it for the purposes of this study."

*one thing is clear from looking at the material in the supplements, is that using the Asher et al. (2002) relationship for both CO2 (invasion) and DMS (evasion) is not correct. The Asher et al. (2002) relation is only for invasion. For evasion, there is a separate equation in Asher and Wannninkhof (1998).*

This is also true and we are glad that it has been pointed out. As the bubble term for DMS is small, there will be negligible impact upon our data. To be absolutely correct, we have adjusted our data and the relevant equations in the revised manuscript.

*Line 54: "These processes include ..." Comment: Buoyancy effects are not a process. It might be better in this sentence to say something like "These processes include diffusion, surface renewal, and bubble-mediated transport. In turn, turbulence can be generated by wind stress, wave-induced mixing, buoyancy currents, and wave breaking."*

Change made.

*Line 56: "A variety of theoretical, laboratory, and field ..." Comment: I don't think this sentence is strictly true. My opinion is we have a fairly good understanding of the factors that affect gas exchange from a phenomenological standpoint (the authors list them just a couple of sentences earlier). What we lack is how to determine which of those processes are important under a given set of circumstances.*

Changed sentence to:
"A variety of theoretical, laboratory, and field approaches have been used to study the processes that control air/sea transfer, but we do not yet have a firm understanding of their relative importance under a range of atmospheric and oceanic conditions."

*Line 60: "Gas transfer via bubbles (k_bub) ..." Comment: It would be good to define k_bub here. The point is that there are a couple of different ways to do this, you can go the Memery and Merlivat (Memery, L. and L. Merlivat (1985). "Modeling of the gas flux through bubbles at the air-water interface." Tellus, Ser. B 37: 272-285) approach and use the bulk air-water concentration difference and accept that k_bub for invasion and evasion are different (e.g., I used this approach in Asher et al. (1996, JGR-Oceans)) or you can redefine the air-water concentration difference in terms of how bubbles would affect the equilibrium and have a common k_bub (but then it might get complicated relating k_bub for invasion and evasion) as done by Woolf (1997).*
We have clarified the definition of k_bub here and cover the issue of invasion vs evasion in detail on Line 161 (to address the comment below).

*Line 78: "These measurements typically show DMS gas transfer velocities that are lower and exhibit more linear wind speed dependence than those estimated for CO2 based on dual tracer studies (e.g. Bell et al., 2015; Yang et al., 2011; Goddijn-Murphy et al., 2012)." Comment: I think the authors should be clear here that there are no CO2 measurements from dual-tracer studies. There are DT measurements for SF6/He, which get related to CO2 through diffusivity. Then there are EC measurements for CO2. Comparison of the DT-derived CO2 transfer velocities with CO2 transfer velocities produced by EC measurements of CO2 fluxes shows relatively good agreement. It is the transfer velocities produced by EC measurements of DMS fluxes that show different behavior.*
Change made.

*Line 87: Comment: maybe want to note that they agree when normalized to a common diffusivity.*
Change made.

*Line 126: "The air side gas transfer contributes about 5% on average to the total resistance for DMS." Comment: The air-side resistance fraction is a function of wind speed. Does this 5% increase as U increases? COAREG must reproduce this, it was measured by McGillis et al. a while back.*
McGillis et al.. (2000) used a non-linear relationship between waterside transfer velocity and wind speed. Based on more recent measurements of DMS gas transfer velocity it is more appropriate to assume a linear relationship.  As a result, the relative contribution of airside resistance to total resistance does not change substantially as a function of wind speed (see Supplemental material in Bell et al 2015).

*Line 161: the relation in the text showing k_w = k_int + k_bub. Comment: I wonder if maybe it is time to stop writing this as a general expression (I know, I am guilty of this as well). What is generally true is that the total gas flux is equal to the sum of interfacial flux and the bubble flux. Saying the overall transfer velocity is equal to the sum of the two transfer velocities really only works if the concentration difference is far from equilibrium. Work through David's relations from the 1997 paper and you'll find they are a bit convoluted in terms of how exactly the pieced (his Delta term) fit together to make a coherent physical picture. If you start by assuming it is the fluxes, not the transfer velocities, which sum linearly, the assumptions required to get to the various relations proposed are more easily understood.*
We have changed the text to be more accurate.

*Technical Comments:*
*1. Multiple citations are not in any recognizable order. Sometimes they are chronological, sometimes alphabetical. I don't remember what the ACP style guide says, but I am sure it is not "random."*
Agreed! We will correct this.

*2. Line 74: "... studies indicate a non-linear dependence ..."*
Changed to "studies observed a non-linear wind speed dependence"

*Line 91: Shouldn't cite papers that are not published or submitted.*
OK, removed.

*Line 175: The two f values are opposite from what is given in the supplement. Not sure which is correct, but it should be consistent (and correct).*
Good spot. Changed in the supplemental material.

*Line 323: I know this is petty, but I don't think Woolf (1997) is based on laboratory data.*
Woolf (1997) did use laboratory data from other studies. We will clarify this.

*Equation 7: Figure caption says "cubic" and equation 7 is quadratic. Resolve this difference.*
Changed.

*Line 384: The citation to Asher and Wanninkhof (1998) should be to Asher et al. (1996). If you really must cite Asher and Wanninkhof (1998) in this context, which you shouldn't, at least make it the other Asher and Wanninkhof (1998) paper that is directly relevant (see citation above).*
Changed.

**Reviewer 2: Ian Brooks**

*Line 86-87: "In that study, no statistically significant difference was observed in gas transfer-wind speed relationships of CO2 and DMS for winds below 10 m s-1" – need to clarify this statement, was a significant difference found for winds above 10 m s-1? Was 10 m s-1 the maximum wind speed in the study?*
Changed to:
"In that study, no data were collected for winds greater than 10 m s-1 and no statistically significant difference was observed in the CO2 and DMS gas transfer-wind speed relationships."

*Line 104-105: as noted in the comment from Blomquist, the air-side resistance is not a function of solubility (though its contribution to the derivation of waterside transfer velocity is dependent on solubility).*
This has been corrected.

*Line 122-123: The use of the COAREG 3.1 model to calculate air-side transfer velocities in order to derive the waterside transfer velocity introduces an assumption that COAREG is providing valid values of ka. Any uncertainty in this will impact the later results and should be acknowledged and if possible quantified.*
We have added the following sentence:
"Note that the use of the COAREG 3.1 model introduces a small uncertainty in our estimates of waterside DMS gas transfer velocity (approximately ±2% when wind speed = 20 m s$^{-1}$)."

*Line 126-127: "The air side gas transfer contributes about 5% on average to the total resistance for DMS" – do you mean 'air-side resistance' rather than air-side transfer?*
Yes. Changed.

*Line 140: "...(Equation 4)..." should be "...(Equation 3)..."*
*Line 170: "...into Equation 6 yields..." should be "...into Equation 5 yields..."*
Changed.

*Line 192 - figures: Figures 1 and 2 are introduced here, but figure 2 is not actually discussed until after discussion of figure 4 breaking the flow of discussion and figures, and leaving me, initially, confused as to how I'd missed the discussion of figure 2. In fact, only the general environmental conditions shown in figure 1 are discussed here, not the gas flux results, which are discussed much later. Figure 1 would be better split into 2, separating the gas fluxes into a figure matching the format of current figure 2. The figures showing the gas flux results could then be placed in a logical order within the discussion. Since wave state is a relevant parameter in the later discussion, it would be useful to add a time series of at least significant wave height to figure 1.*

We agree. Figure 1 has been split and a timeseries of Hs has been added. Figures are reordered so the gas flux/transfer velocity figures follow the whitecapping figures.

*Line 207: The authors note that their estimates of whitecap fraction as a function of wind speed are substantially lower than other recently published values – at times an order of magnitude lower. A likely reason for this is the exposure settings on the camera. During the HiWinGS project cruise in 2013 two independent sets of cameras were used for whitecap imaging. They were initially found to give whitecap fractions that differed by a factor of several. Tests were conducted during the final transit of the cruise, in which a pair of identical cameras were run side by side; one with fixed exposure settings, the other having the exposure settings changed every few hours. The exposure settings were found to make a substantial difference to the whitecap fraction calculated using the same Callaghan and White (2009) algorithm used here – up to a factor of 4 for the range of settings tested. It was found that almost all of this difference (both between the 2 cameras in the exposure trial, and between the two sets of cameras used throughout the cruise) was removed if the images were first 'normalised' to remove any brightness gradient across the image. Brief details of these tests will be given in Brumer, S. E., C. J. Zappa, I. M. Brooks, H. Tamura, S. M. Brown, B. Blomquist, C. W. Fairall, A. Cifuentes-Lorenzen, 2017: Whitecap coverage dependence on wind and wave statistics as observed during SO GasEx and HiWinGS, J. Phys. Oceanogr. (under revision)*

Many of the potential issues highlighted above (exposure settings, intercomparability of image processing) have actually been addressed. We have added additional information about the whitecap image processing to the Supplemental material.

The lower bounds of the Knorr $W$ data do fall below recent parameterisations, most notably in the wind speed range of ~ 7.5 m s$^{-1}$ to 12.5 m s$^{-1}$. However, when the Knorr $W$ data are binned by wind speed (squares in Figure 3a, main manuscript), the binned $W$ data compare favourably with the Schwendeman and Thomson (2015) estimate of whitecap fraction at all wind speeds, with the exception of the lowest bin of 3 m s$^{-1}$. The binned Knorr $W$ data also agree favourably with Callaghan et al., (2008), except for the 9 m s$^{-1}$ and 11 m s$^{-1}$ binned datapoints, for which the Knorr binned $W$ data are lower. When binned by wind speed, the Knorr $W$ data and the two parameterisations generally fall within a factor of 2 across the wind speeds examined.

There is clearly quite a lot of scatter in the Knorr $W$ dataset, and many data points lie below the Schwendeman and Thomson (2015) and Callaghan et al., (2008) parameterisations. However, we do not believe that the primary driver of the differences observed is due to the image processing methodology employed. Rather, it could be that other sources of variability (at a given wind speed) have caused the observed differences. Examples include: (i) water chemistry (surfactants); (ii) total wave field energy dissipation; and (iii) energy dissipation by microscale breaking waves as opposed to air-entraining whitecaps. It is beyond the scope of this paper to address these potential causes.

*Line 213-215: "Stage A whitecap fraction data is highly variable at ~11 m s-1 213 wind speeds (Figure 3b), which is driven by the difference in the wind-wave conditions during Knorr_11 (ST184 vs ST191, Figure 4a)" – two points:*
*(1) the difference is ascribed to different wind-wave conditions at the two stations, but no wave data are shown. As noted above, relevant wave parameters need to be added to figure 1.*
Done.

*(2) A similar broad range of stage A whitecaps is evident at around U = 6 m s-1, also resulting from grouping of high/low values by different stations...are the wind-wave conditions similarly different in this case?*
We agree that there is a grouping at U=6 m s$^{-1}$. It is unclear what caused this (although ST181 was in the Gulf Stream and thus had much higher water temperatures). However, this discussion is beyond the scope of this paper.

*Lines 218-223: The authors first note that where stage A whitecap fractions is < 10-4 the relationship with RH is more scattered than at higher fractions; they then note a number of factors that affect wave breaking and so*

*whitecap fraction, but don't make a coherent link back to their initial point about the scatter in the stage-A whitecap / RH relationship. This reads as an almost unconnected series of statements...all true, but leaving the reader wondering what the point being made is.*

We agree that this text needed adjusting. We also decided that the text discussing Stage A whitecap variability and $R_H$ was not essential to the manuscript and have moved the modified text and the related Figure to Supplemental information.

*Line 282: "...(Figure 5)" -> "...(Figure 5b,d)" – again text refers to high wave conditions for ST191 but no wave data provided for reader to assess.*

In this instance, we feel that it is sufficient for us to reference the in depth discussion in Bell et al. (2013).

*Figure 5 and the discussion of it have some general issues:*
*- It's hard to see the pink/green lines against the mass of pink/green dots on panels a and b – it would help here to plot the dots in a paler shade of pink/green to allow the lines to stand out.*

The figure has been adjusted to improve readability.

*- The curves shown, for the COAREG3.1 model are a useful reference, but fits to the actual data are also needed; these would allow a much clearer assessment of how closely the COARE model agrees with the observations.*

We tried this but it made the plots too busy. We decided it best not to include both.

*- lines 282-284: "Under the high wind, high wave conditions encountered during ST191, the wind speed-dependence of kDMS,Sc was lower than expected, with a slope roughly half that of the rest of the cruise data. This effect was not observed at ST184." – since the ST191 data are not highlighted in any way it is not possible for the reader to judge the behaviour here. Of note perhaps is not simply the high wind and wave conditions during ST191 but the different time history of the winds – a sustained period of high winds during ST191 vs a very short period in ST184 where the wind rises rapidly, spikes, and decreases rapidly. These two periods are likely to produce very different wave fields at the same wind speeds – again, reason to plot the wave parameters in figure 1 – which might explain the very different whitecap fractions seen in Figure 1b for these periods.*

In addition to plotting the wave data, we have added the word 'sustained' to this sentence and referenced Bell et al. 2013)

*Line 294 & 305: the phrasing "until 11 m s-1 wind speed" is rather clumsy; 'until' implies a variation over time, which is not what is meant – "...up to wind speeds of 11 m s-1" would read better.*

Change made.

*Line 326: "...Δkw is near zero at very low wind speeds (U10 ≤ 4.5 m s-1 )..." – this is hard to judge. Eyeballing the data points I would agree; however, there are only 3 points at U < 4.5 m s-1, and all of those at U>3 m s-1. Their mean Δk is ~5 cm/hr and the fitted curve approaches a Δkw of ~3 cm/hr at U = 0, not zero. One might argue for an alternative functional form in which the exponents were not prescribed might better represent the data; the quadratic used here implicitly assumes the functional dependence.*

We have tested a variety of functional forms and number of exponents for our best fit: linear, polynomial (n=2,3), power (n=1,2) and exponential (n=1,2). The goodness of fit was extremely comparable between all fits (from $R^2 = 0.66$ to $R^2 = 0.67$) with the exception of the linear fit ($R^2 = 0.62$). As the goodness of fit did not help with our decision, the choice of fit is subjective. Having considered the different fit lines, we feel that a simple power law fit ($\Delta k_w = 0.177 U_{10}^{1.928}$) represents the low $\Delta k_w$ data better than our original choice of fit. We have changed the manuscript accordingly.

The fit line to the data tells us that $\Delta k_w$ is indeed positive at low wind speeds. This could be driven by effects such as chemical enhancement and sea surface skin temperature (see Discussion), which we have not controlled for here. We used the term 'near zero' intentionally but now clarify this in the revised manuscript with "($< 4.5$ cm/hr)"

*What is the reasoning behind using 4-hour averages of transfer velocities here (and elsewhere)? 4 hours is quite a long time relative to the time period over which significant changes in forcing can take place. Granted it greatly reduces scatter, but I would wary of averaging over periods much long than ~1 hour.*
Some averaging of the data was necessary to reduce the scatter, but we accept that minimising the averaging time period is important. We have changed our averaging period to two hours.

*Line 358: "...the relationship between Δkw and whitecap areal extent appears to be linear." – I'm not convinced this is entirely true – approximately so over the range 0.005 < W < 0.05, but ~half the data points lie at W < 0.005, and seem to drop off rather more rapidly than the fit to the high values of W would indicate. Plotting W on a log scale might give a rather different impression. Not that IF the relationship is linear as suggested then eyeballing a fit (if you claim a linear fit, it would help to show it!) suggests Δk = 10-15 cm/hr at W = 0, which raises questions as to why that should be when there are no bubbles to account for a difference in k, and why this minimum difference is several times higher than that derived as a function of wind speed. If on the other hand a roughly linear fit of Δk to log(W) existed (which I think the rapid drop off in Δk at very low W might support) then Δk would approach zero at low W.*
We have revised the statement to:
"The functional form of the relationship between $\Delta k_w$ and whitecap areal extent appears to be linear for $W_T >$ 0.005. However, the Knorr_11 dataset is small and quite scattered, particularly when $W_T < 0.005$."

*Line 366-367: "In this case, Δkw should be more strongly correlated with WA than WB or WT." – in a general sense, this is true, but is only if the various factors affecting foam decay vary. If foam decay rate is constant then WT should be proportional to WA.*
We agree. We feel this case is broadly covered later in the paragraph. We have adjusted the text slightly to take into account a constant decay rate:
"$W_T$ and $W_A$ may be equally good (or poor) proxies for bubbles because: (i) surfactant activity was either insignificant or sufficiently invariant in the study region (despite high biological productivity) that $W_B$ does not confound the relationship between $W_T$ and $W_A$"

*Line 387: "Both models significantly underestimate kbub,CO2 at wind speeds below about 11 m s-1." Actually both models rather underestimate the observationally derived Kbub,CO2 at all windspeeds; however, note that both models are driven by the observed wind-whitecap relationship, which has already been stated to be low compared to other recent estimates. Is the agreement better using a whitecap function that agrees more closely with the recent consensus?*
The agreement is slightly improved (at intermediate wind speeds) using a different whitecap function. Although we see no reason to doubt our whitecap measurements, it is useful to observe the importance of the wind speed-whitecap fraction relationship to the output from these models, so we have added these $k_{bub,CO2}$ estimates to the manuscript.

*While the Asher et al. model is lower than the observations, it is not wildly so (essentially matching the lower boundary of the observed values), and the agreement in both values and functional behaviour is rather convincing.*
We are possibly less optimistic on this point and hopefully can agree to disagree!

*Line 455: "...eddy covariance setup..." -> "...eddy covariance system..."*
Change made.

*Line 463: reference to paper in preparation not generally allowed...if it's not 'in press' by the time this manuscript it copy edited, the copy editor will want the reference cut.*
Change made.

*Figure 3: it's hard to pick out the curves and black square points against the mass of black dots.*

*Suggest changing black dots to mid-grey and ensuring everything else is plotted over them. It would also be good to see a functional fit to this data as well as the functions from previous studies...especially as this function is later used to drive the kbub,co2 models plotted in figure 8.* Revised accordingly.